# ACCELERATED GRADIENT FLOW FOR PROBABILITY DISTRIBUTIONS

## ABSTRACT

This paper presents a methodology and numerical algorithms for constructing accelerated gradient flows on the space of probability distributions. In particular, we extend the recent variational formulation of accelerated gradient methods in Wibisono et al. (2016) from vector valued variables to probability distributions. The variational problem is modeled as a mean-field optimal control problem. The maximum principle of optimal control theory is used to derive Hamilton's equations for the optimal gradient flow. The Hamilton's equation are shown to achieve the accelerated form of density transport from any initial probability distribution to a target probability distribution. A quantitative estimate on the asymptotic convergence rate is provided based on a Lyapunov function construction, when the objective functional is displacement convex. Two numerical approximations are presented to implement the Hamilton's equations as a system of $N$ interacting particles. The continuous limit of the Nesterov's algorithm is shown to be a special case with $N = 1$. The algorithm is illustrated with numerical examples.

## 1   INTRODUCTION

Optimization on the space of probability distributions is important to a number of machine learning models including variational inference (Blei et al., 2017), generative models (Goodfellow et al., 2014; Arjovsky et al., 2017), and policy optimization in reinforcement learning (Sutton et al., 2000). A number of recent studies have considered solution approaches to these problems based upon a construction of gradient flow on the space of probability distributions (Zhang et al., 2018; Liu & Wang, 2016; Frogner & Poggio, 2018; Chizat & Bach, 2018; Richemond & Maginnis, 2017; Chen et al., 2018). Such constructions are useful for convergence analysis as well as development of numerical algorithms.

In this paper, we propose a methodology and numerical algorithms that achieve accelerated gradient flows on the space of probability distributions. The proposed numerical algorithms are related to yet distinct from the accelerated stochastic gradient descent (Jain et al., 2017) and Hamiltonian Markov chain Monte-Carlo (MCMC) algorithms (Neal et al., 2011; Cheng et al., 2017). The proposed methodology extends the variational formulation of (Wibisono et al., 2016) from vector valued variables to probability distributions. The original formulation of Wibisono et al. (2016) was used to derive and analyze the convergence properties of a large class of accelerated optimization algorithms, most significant of which is the continuous-time limit of the Nesterov's algorithm (Su et al., 2014). In this paper, the limit is referred to as the Nesterov's ordinary differential equation (ODE).

The extension proposed in our work is based upon a generalization of the formula for the Lagrangian in Wibisono et al. (2016): (i) the kinetic energy term is replaced with the expected value of kinetic energy; and (ii) the potential energy term is replaced with a suitably defined functional on the space of probability distributions. The variational problem is to obtain a trajectory in the space of probability distributions that minimizes the action integral of the Lagrangian.

The variational problem is modeled as a mean-field optimal problem. The maximum principle of the optimal control theory is used to derive the Hamilton's equations which represent the first order optimality conditions. The Hamilton's equations provide a generalization of the Nesterov's ODE to the space of probability distributions. A candidate Lyapunov function is proposed for the convergence analysis of the solution of the Hamilton's equations. In this way, quantitative estimates on convergence rate are obtained for the case when the objective functional is displacement convex (McCann, 1997). Table 1 provides a summary of the relationship between the original variational formulation in Wibisono et al. (2016) and the extension proposed in this paper.

We also consider the important special case when the objective functional is the relative entropy functional $D(\rho|\rho_\infty)$ defined with respect to a target probability distribution $\rho_\infty$. In this case, the accelerated gradient flow is shown to be related to the continuous limit of the Hamiltonian Monte-Carlo algorithm (Cheng et al., 2017) (Remark 1). The Hamilton's equations are finite-dimensional for the special case when the initial and the target probability distributions are both Gaussian. In this case, the mean evolves according to the Nesterov's ODE. For the general case, the Lyapunov function-based convergence analysis applies when the target distribution is log-concave.

As a final contribution, the proposed methodology is used to obtain a numerical algorithm. The algorithm is an interacting particle system that empirically approximates the distribution with a finite but large number of $N$ particles. The difficult part of this construction is the approximation of the interaction term between particles. For this purpose, two types of approximations are described: (i) Gaussian approximation which is asymptotically (as $N \to \infty$) exact in Gaussian settings; and (ii) Diffusion map approximation which is computationally more demanding but asymptotically exact for a general class of distributions.

The outline of the remainder of this paper is as follows: Sec. 2 provides a brief review of the variational formulation in Wibisono et al. (2016). The proposed extension to the space of probability distribution appears in Sec. 3 where the main result is also described. The numerical algorithm along with the results of numerical experiments appear in Sec. 4. Comparisons with MCMC and Hamiltonian MCMC are also described. The conclusions appear in Sec. 5.

**Notation:** The gradient and divergence operators are denoted as $\nabla$ and $\nabla\cdot$ respectively. With multiple variables, $\nabla_z$ denotes the gradient with respect to the variable $z$. Therefore, the divergence of the vector field $U$ is $\nabla \cdot U(x) = \sum_{n=1}^d \nabla_{x_n} U_n(x)$. The space of absolutely continuous probability measures on $\mathbb{R}^d$ with finite second moments is denoted by $\mathcal{P}_{\text{ac},2}(\mathbb{R}^d)$. The Wasserstein gradient and the Gâteaux derivative of a functional $F$ is denoted as $\nabla_\rho F(\rho)$ and $\frac{\partial F}{\partial \rho}(\rho)$ respectively (see Appendix C for definition). The probability distribution of a random variable $Z$ is denoted as $\text{Law}(Z)$.

| | Vector | Probability distribution |
|---|---|---|
| State-space | $\mathbb{R}^d$ | $\mathcal{P}_2(\mathbb{R}^d)$ |
| Objective function | $f(x)$ | $\mathsf{F}(\rho) := \mathsf{D}(\rho\|\rho_\infty)$ |
| Lagrangian | $e^{\alpha_t+\gamma_t}\left(\frac{1}{2}|e^{-\alpha_t}u|^2 - e^{\beta_t}f(x)\right)$ | $e^{\alpha_t+\gamma_t}\mathsf{E}\left[\frac{1}{2}|e^{-\alpha_t}U|^2 - e^{\beta_t}\log(\frac{\rho(X)}{\rho_\infty(X)})\right]$ |
| Lyapunov funct. | $\frac{1}{2}\left|x+e^{-\gamma_t}y-\bar{x}\right|^2$ $+e^{\beta_t}(f(x)-f(\bar{x}))$ | $\frac{1}{2}\mathsf{E}[|X_t+e^{-\gamma_t}Y_t-T^{\rho_\infty}_{\rho_t}(X_t)|^2]$ $+e^{\beta_t}(\mathsf{F}(\rho_t)-\mathsf{F}(\rho_\infty))$ |

Table 1: Summary of the variational formulations for vectors and probability distributions.

## 2 REVIEW OF THE VARIATIONAL FORMULATION OF WIBISONO ET AL. (2016)

The basic problem is to minimize a $C^1$ smooth convex function $f$ on $\mathbb{R}^d$. The standard form of the gradient descent algorithm for this problem is an ODE:

$$\frac{\mathrm{d}X_t}{\mathrm{d}t} = -\nabla f(X_t), \quad t \geq 0 \tag{1}$$

Accelerated forms of this algorithm are obtained based on a variational formulation due to Wibisono et al. (2016). The formulation is briefly reviewed here using an optimal control formalism. The Lagrangian $L : \mathbb{R}^+ \times \mathbb{R}^d \times \mathbb{R}^d \to \mathbb{R}$ is defined as

$$L(t,x,u) := e^{\alpha_t+\gamma_t}\left(\underbrace{\frac{1}{2}|e^{-\alpha_t}u|^2}_{\text{kinetic energy}} - \underbrace{e^{\beta_t}f(x)}_{\text{potential energy}}\right) \tag{2}$$

where $t \geq 0$ is the time, $x \in \mathbb{R}^d$ is the state, $u \in \mathbb{R}^d$ is the velocity or control input, and the time-varying parameters $\alpha_t, \beta_t, \gamma_t$ satisfy the following scaling conditions: $\alpha_t = \log p - \log t$, $\beta_t = p\log t + \log C$, and $\gamma_t = p\log t$ where $p \geq 2$ and $C > 0$ are constants.

The variational problem is

$$\begin{aligned}\underset{u}{\text{Minimize}} \quad & J(u) = \int_0^\infty L(t,X_t,u_t)\,\mathrm{d}t \\ \text{Subject to} \quad & \frac{\mathrm{d}X_t}{\mathrm{d}t} = u_t, \quad X_0 = x_0\end{aligned} \tag{3}$$

The Hamiltonian function is

$$H(t,x,y,u) = y \cdot u - L(t,x,u) \tag{4}$$

where $y \in \mathbb{R}^d$ is dual variable and $y \cdot u$ denotes the dot product between vectors $y$ and $u$.

According to the Pontryagin's Maximum Principle, the optimal control $u_t^* = \underset{v}{\arg\max}\, H(t,X_t,Y_t,v) = e^{\alpha_t-\gamma_t}Y_t$. The resulting Hamilton's equations are

$$\frac{\mathrm{d}X_t}{\mathrm{d}t} = +\nabla_y H(t,X_t,Y_t,u_t) = e^{\alpha_t-\gamma_t}Y_t, \quad X_0 = x_0 \tag{5a}$$

$$\frac{\mathrm{d}Y_t}{\mathrm{d}t} = -\nabla_x H(t,X_t,Y_t,u_t) = -e^{\alpha_t+\beta_t+\gamma_t}\nabla f(X_t), \quad Y_0 = y_0 \tag{5b}$$

The system (5) is an example of accelerated gradient descent algorithm. Specifically, if the parameters $\alpha_t, \beta_t, \gamma_t$ are defined using $p = 2$, one obtains the continuous-time limit of the Nesterov's accelerated algorithm. It is referred to as the Nesterov's ODE in this paper.

For this system, a Lyapunov function is as follows:

$$V(t,x,y) = \frac{1}{2}\left|x+e^{-\gamma_t}y-\bar{x}\right|^2 + e^{\beta_t}(f(x)-f(\bar{x})) \tag{6}$$

where $\bar{x} \in \arg\min_x f(x)$. It is shown in Wibisono et al. (2016) that upon differentiating along the solution trajectory, $\frac{\mathrm{d}}{\mathrm{d}t} V(t, X_t, Y_t) \leq 0$. This yields the following convergence rate:

$$f(X_t) - f(\bar{x}) \leq O(e^{-\beta_t}), \quad \forall t \geq 0 \tag{7}$$

## 3    VARIATIONAL FORMULATION FOR PROBABILITY DISTRIBUTIONS

### 3.1    MOTIVATION AND BACKGROUND

Let $\mathsf{F} : \mathcal{P}_{\mathrm{ac},2}(\mathbb{R}^d) \to \mathbb{R}$ be a functional on the space of probability distributions. Consider the problem of minimizing $\mathsf{F}(\rho)$. The (Wasserstein) gradient flow with respect to $\mathsf{F}(\rho)$ is

$$\frac{\partial \rho_t}{\partial t} = \nabla \cdot (\rho_t \nabla_\rho \mathsf{F}(\rho_t)) \tag{8}$$

where $\nabla_\rho \mathsf{F}(\rho)$ is the Wasserstein gradient of $\mathsf{F}$.

An important example is the relative entropy functional where $\mathsf{F}(\rho) = \mathsf{D}(\rho|\rho_\infty) := \int_{\mathbb{R}^d} \log(\frac{\rho(x)}{\rho_\infty(x)}) \rho(x) \, \mathrm{d}x$ where $\rho_\infty \in \mathcal{P}_{\mathrm{ac},2}(\mathbb{R}^d)$ is referred to as the target distribution. The gradient of relative entropy is given by $\nabla_\rho \mathsf{F}(\rho) = \nabla \log(\frac{\rho}{\rho_\infty})$. The gradient flow

$$\frac{\partial \rho_t}{\partial t} = -\nabla \cdot (\rho_t \nabla \log(\rho_\infty)) + \Delta \rho_t \tag{9}$$

is the Fokker-Planck equation (Jordan et al., 1998). The gradient flow achieves the density transport from an initial probability distribution $\rho_0$ to the target (here, also equilibrium) probability distribution $\rho_\infty$; and underlies the construction and the analysis of Markov chain Monte-Carlo (MCMC) algorithms. The simplest MCMC algorithm is the Langevin stochastic differential equation (SDE):

$$\mathrm{d}X_t = -\nabla f(X_t) \, \mathrm{d}t + \sqrt{2} \, \mathrm{d}B_t, \quad X_0 \sim \rho_0$$

where $B_t$ is the standard Brownian motion in $\mathbb{R}^d$.

The main problem of this paper is to construct an accelerated form of the gradient flow (8). The proposed solution is based upon a variational formulation. As tabulated in Table 1, the solution represents a generalization of Wibisono et al. (2016) from its original deterministic finite-dimensional to now probabilistic infinite-dimensional settings.

The variational problem can be expressed in two equivalent forms: (i) The probabilistic form is described next in the main body of the paper; and (ii) The partial differential equation (PDE) form appears in the Appendix. The probabilistic form is stressed here because it represents a direct generalization of the Nesterov's ODE and because it is closer to the numerical algorithm.

### 3.2    PROBABILISTIC FORM OF THE VARIATIONAL PROBLEM

Consider the stochastic process $\{X_t\}_{t \geq 0}$ that takes values in $\mathbb{R}^d$ and evolves according to:

$$\frac{\mathrm{d}X_t}{\mathrm{d}t} = U_t, \quad X_0 \sim \rho_0$$

where the control input $\{U_t\}_{t \geq 0}$ also takes values in $\mathbb{R}^d$, and $\rho_0 \in \mathcal{P}_{\mathrm{ac},2}(\mathbb{R}^d)$ is the probability distribution of the initial condition $X_0$. It is noted that the randomness here comes only from the random initial condition.

Suppose the objective functional is of the form $\mathsf{F}(\rho) = \int \tilde{F}(\rho, x) \rho(x) \, \mathrm{d}x$. The Lagrangian $\mathsf{L} : \mathbb{R}^+ \times \mathbb{R}^d \times \mathcal{P}_{\mathrm{ac},2}(\mathbb{R}^d) \times \mathbb{R}^d \to \mathbb{R}$ is defined as

$$\mathsf{L}(t, x, \rho, u) := e^{\alpha_t + \gamma_t} \left( \underbrace{\frac{1}{2} |e^{-\alpha_t} u|^2}_{\text{kinetic energy}} - \underbrace{e^{\beta_t} \tilde{F}(\rho, x)}_{\text{potential energy}} \right) \tag{10}$$

This formula is a natural generalization of the Lagrangian (2) and the parameters $\alpha_t, \beta_t, \gamma_t$ are defined exactly the same as in the finite-dimensional case. The stochastic optimal control problem is:

$$\text{Minimize} \quad \mathsf{J}(u) = \mathsf{E}\left[\int_0^\infty \mathsf{L}(t, X_t, \rho_t, U_t)\, \mathrm{d}t\right]$$

$$\text{Subject to} \quad \frac{\mathrm{d}X_t}{\mathrm{d}t} = U_t, \quad X_0 \sim \rho_0 \tag{11}$$

where $\rho_t = \mathrm{Law}(X_t) \in \mathcal{P}_{\mathrm{ac},2}(\mathbb{R}^d)$ is the probability density function of the random variable $X_t$.

The Hamiltonian function $\mathsf{H} : \mathbb{R}^+ \times \mathbb{R}^d \times \mathcal{P}_{\mathrm{ac},2}(\mathbb{R}^d) \times \mathbb{R}^d \times \mathbb{R}^d \to \mathbb{R}$ for this problem is given by (Carmona & Delarue, 2017, Sec. 6.2.3):

$$\mathsf{H}(t, x, \rho, y, u) := u \cdot y - \mathsf{L}(t, x, \rho, u) \tag{12}$$

where $y \in \mathbb{R}^d$ is the dual variable.

### 3.3 MAIN RESULT

**Theorem 1.** *Consider the variational problem* (11).

(i) *The optimal control* $U_t^* = e^{\alpha_t - \gamma_t} Y_t$ *where the optimal trajectory* $\{(X_t, Y_t)\}_{t \geq 0}$ *evolves according to the Hamilton's equations:*

$$\frac{\mathrm{d}X_t}{\mathrm{d}t} = U_t^* = e^{\alpha_t - \gamma_t} Y_t, \quad X_0 \sim \rho_0 \tag{13a}$$

$$\frac{\mathrm{d}Y_t}{\mathrm{d}t} = -e^{\alpha_t + \beta_t + \gamma_t} \nabla_\rho \mathsf{F}(\rho_t)(X_t), \quad Y_0 = \nabla\phi_0(X_0) \tag{13b}$$

*where $\phi_0$ is any convex function and $\rho_t := Law(X_t)$.*

(ii) *Suppose also that the functional $\mathsf{F}$ is displacement convex and $\rho_\infty$ is its minimizer. Define the energy along the optimal trajectory*

$$V(t) = \frac{1}{2}\mathsf{E}[|X_t + e^{-\gamma_t} Y_t - T_{\rho_t}^{\rho_\infty}(X_t)|^2] + e^{\beta_t}(\mathsf{F}(\rho) - \mathsf{F}(\rho_\infty)) \tag{14}$$

*where the map $T_{\rho_t}^{\rho_\infty} : \mathbb{R}^d \to \mathbb{R}^d$ is the optimal transport map from $\rho_t$ to $\rho_\infty$. Suppose also that the following technical assumption holds: $\mathsf{E}[(X_t + e^{-\gamma_t} Y_t - T_{\rho_t}^{\rho_\infty}(X_t)) \cdot \frac{\mathrm{d}}{\mathrm{d}t} T_{\rho_t}^{\rho_\infty}(X_t)] = 0$. Then $\frac{\mathrm{d}V}{\mathrm{d}t}(t) \leq 0$. Consequently, the following rate of convergence is obtained along the optimal trajectory:*

$$\mathsf{F}(\rho_t) - \mathsf{F}(\rho_\infty) \leq O(e^{-\beta_t}), \quad \forall t \geq 0 \tag{15}$$

*Proof sketch.* The Hamilton's equations are derived using the standard mean-field optimal control theory Carmona & Delarue (2017). The Lyapunov function argument is based upon the variational inequality characterization of a displacement convex function (Ambrosio et al., 2008, Eq. 10.1.7). The detailed proof appears in the Appendix. We expect that the technical assumption is not necessary. This is the subject of the continuing work. $\qquad\square$

### 3.4 RELATIVE ENTROPY AS THE FUNCTIONAL

In the remainder of this paper, we assume that the functional $\mathsf{F}(\rho) = \mathsf{D}(\rho|\rho_\infty)$ is the relative entropy where $\rho_\infty \in \mathcal{P}_{\mathrm{ac},2}(\mathbb{R}^d)$ is a given target probability distribution. In this case the Hamilton's equations are given by

$$\frac{\mathrm{d}X_t}{\mathrm{d}t} = e^{\alpha_t - \gamma_t} Y_t, \quad X_0 \sim \rho_0 \tag{16a}$$

$$\frac{\mathrm{d}Y_t}{\mathrm{d}t} = -e^{\alpha_t + \beta_t + \gamma_t}(\nabla f(X_t) + \nabla \log(\rho_t(X_t)), \quad Y_0 = \nabla\phi_0(X_0) \tag{16b}$$

where $\rho_t = \text{Law}(X_t)$ and $f = -\log(\rho_\infty)$. Moreover, if $f$ is convex (or equivalently $\rho_\infty$ is log-concave), then F is displacement convex with the unique minimizer at $\rho_\infty$ and the convergence estimate is given by $\mathsf{D}(\rho_t|\rho_\infty) \leq O(e^{-\beta_t})$.

**Remark 1.** *The Hamilton's equations* (16) *with the relative entropy functional is related to the under-damped Langevin equation (Cheng et al., 2017). The difference is that the deterministic term* $\nabla \log(\rho_t)$ *in* (16) *is replaced with a random Brownian motion term in the under-damped Langevin equation. More detailed comparison appears in the Appendix D.*

### 3.5 QUADRATIC GAUSSIAN CASE

Suppose the initial distribution $\rho_0$ and the target distribution $\rho_\infty$ are both Gaussian, denoted as $\mathcal{N}(m_0, \Sigma_0)$ and $\mathcal{N}(\bar{x}, Q)$, respectively. This is equivalent to the objective function $f(x)$ being quadratic of the form $f(x) = \frac{1}{2}(x - \bar{x})^\top Q^{-1}(x - \bar{x})$. Therefore, this problem is referred to as the *quadratic Gaussian case*. The following Proposition shows that the mean of the stochastic process $(X_t, Y_t)$ evolves according to the Nesterov ODE (5):

**Proposition 1.** *(Quadratic Gaussian case) Consider the variational problem* (11) *for the quadratic Gaussian case. Then*

(i) *The stochastic process $(X_t, Y_t)$ is a Gaussian process. The Hamilton's equations are given by:*

$$\frac{\mathrm{d}X_t}{\mathrm{d}t} = e^{\alpha_t - \gamma_t} Y_t, \quad \frac{\mathrm{d}Y_t}{\mathrm{d}t} = -e^{\alpha_t + \beta_t + \gamma_t}(Q^{-1}(X_t - \bar{x}) - \Sigma_t^{-1}(X_t - m_t))$$

*where $m_t$ and $\Sigma_t$ are the mean and the covariance of $X_t$.*

(ii) *Upon taking the expectation of both sides, and denoting $n_t := \mathsf{E}[Y_t]$*

$$\frac{\mathrm{d}m_t}{\mathrm{d}t} = e^{\alpha_t - \gamma_t} n_t, \quad \frac{\mathrm{d}n_t}{\mathrm{d}t} = -e^{\alpha_t + \beta_t + \gamma_t} \underbrace{Q^{-1}(m_t - \bar{x})}_{\nabla f(m_t)}$$

*which is identical to Nesterov ODE (5).*

## 4 NUMERICAL ALGORITHM

The proposed numerical algorithm is based upon an interacting particle implementation of the Hamilton's equation (16). Consider a system of $N$ particles $\{(X_t^i, Y_t^i)\}_{i=1}^N$ that evolve according to:

$$\frac{\mathrm{d}X_t^i}{\mathrm{d}t} = e^{\alpha_t - \gamma_t} Y_t^i, \quad X_0^i \sim \rho_0$$

$$\frac{\mathrm{d}Y_t^i}{\mathrm{d}t} = -e^{\alpha_t + \beta_t + \gamma_t}(\nabla f(X_t^i) + \underbrace{I_t^{(N)}(X_t^i)}_{\text{interaction term}}), \quad Y_0^i = \nabla \phi_0(X_0^i)$$

The interaction term $I_t^{(N)}$ is an empirical approximation of the $\nabla \log(\rho_t)$ term in (16). We propose two types of empirical approximations as follows:

**1. Gaussian approximation:** Suppose the density is approximated as a Gaussian $\mathcal{N}(m_t, \Sigma_t)$. In this case, $\nabla \log(\rho_t(x)) = -\Sigma_t^{-1}(x - m_t)$. This motivates the following empirical approximation of the interaction term:

$$I_t^{(N)}(x) = -\Sigma_t^{(N)-1}(x - m_t^{(N)}) \tag{18}$$

---

**Algorithm 1** Interacting particle implementation of the accelerated gradient flow

---

**Input:** $\rho_0, \phi_0, N, t_0, \Delta t, p, C, K$

**Output:** $\{X_k^i\}_{i=1,k=0}^{N,K}$

   Initialize $\{X_0^i\}_{i=1}^N \overset{\text{i.i.d}}{\sim} \rho_0$, $Y_0^i = \nabla\phi_0(X_0^i)$

   Compute $I_0^{(N)}(X_0^i)$ with (18) or (19)

   **for** $k = 0$ to $K - 1$ **do**

      $t_{k+\frac{1}{2}} = t_k + \frac{1}{2}\Delta t$

      $Y_{k+\frac{1}{2}}^i = Y_k^i - \frac{1}{2}Cpt_{k+\frac{1}{2}}^{2p-1}(\nabla f(X_k^i) + I_k^{(N)}(X_k^i))\Delta t$

      $X_{k+1}^i = X_k^i + \frac{p}{t_{k+\frac{1}{2}}^{p+1}}Y_k^i\Delta t$

      Compute $I_{k+1}^{(N)}(X_{k+1}^i)$ with (18)or (19)

      $Y_{k+1}^i = Y_{k+\frac{1}{2}}^i - \frac{1}{2}Cpt_{k+\frac{1}{2}}^{2p-1}(\nabla f(X_{k+1}^i) + I_{k+1}^{(N)}(X_{k+1}^i))\Delta t$

      $t_{k+1} = t_{k+\frac{1}{2}} + \frac{1}{2}\Delta t$

   **end for**

---

where $m_t^{(N)} := N^{-1}\sum_{i=1}^N X_t^i$ is the empirical mean and $\Sigma_t^{(N)} := \frac{1}{N-1}\sum_{i=1}^N(X_t^i - m_t^{(N)})(X_t^i - m_t^{(N)})^\top$ is the empirical covariance.

Even though the approximation is asymptotically (as $N \to \infty$) exact only under the Gaussian assumption, it may be used in a more general settings, particularly when the density $\rho_t$ is unimodal. The situation is analogous to the (Bayesian) filtering problem, where an ensemble Kalman filter is used as an approximate solution for non-Gaussian distributions (Evensen, 2003).

**2. Diffusion map approximation:** This is based upon the diffusion map approximation of the weighted Laplacian operator (Coifman & Lafon, 2006; Hein et al., 2007). For a $C^2$ function $f$, the weighted Laplacian is defined as $\Delta_\rho f := \frac{1}{\rho}\nabla \cdot (\rho\nabla f)$. Denote $e(x) = x$ as the coordinate function on $\mathbb{R}^d$. It is a straightforward calculation to show that $\nabla \log(\rho) = \Delta_\rho e$. This allows one to use the diffusion map approximation of the weighted Laplacian to approximate the interaction term as follows:

$$(\text{DM}) \quad I_t^{(N)}(X_t^i) = \frac{1}{\epsilon}\frac{\sum_{j=1}^N k_\epsilon(X_t^i, X_t^j)(X_t^j - X_t^i)}{\sum_{j=1}^N k_\epsilon(X_t^i, X_t^j)} \tag{19}$$

where the kernel $k_\epsilon(x,y) = \frac{g_\epsilon(x,y)}{\sqrt{\sum_{i=1}^N g_\epsilon(y,X^i)}}$ is constructed empirically in terms of the Gaussian kernel $g_\epsilon(x,y) = \exp(-|x-y|^2/(4\epsilon))$. The parameter $\epsilon$ is referred to as the kernel bandwidth. The approximation is asymptotically exact as $\epsilon \downarrow 0$ and $N \uparrow \infty$. The approximation error is of order $O(\epsilon) + O(\frac{1}{\sqrt{N}\epsilon^{d/4}})$ where the first term is referred to as the bias error and the second term is referred to as the variance error (Hein et al., 2007). The variance error is the dominant term in the error for small values of $\epsilon$, whereas the bias error is the dominant term for large values of $\epsilon$ (see Figure 3(d)).

The resulting interacting particle algorithm is tabulated in Table 1. The symplectic method proposed in (Betancourt et al., 2018) is used to carry out the numerical integration. The algorithm is applied to two examples as described in the following sections.

**Remark 2.** *For the case where there is only one particle ( $N = 1$), the interaction term is zero and the system* (17) *reduces to the Nesterov ODE* (5).

**Remark 3.** *(Comparison with density estimation) The diffusion map approximation algorithm is conceptually different from an explicit density estimation-based approach. A basic density estimation is to approximate $\rho(x) \approx \frac{1}{N}\sum_{i=1}^N g_\epsilon(x, X_t^i)$ where $g_\epsilon(x,y)$ is the Gaussian kernel. Using such an*

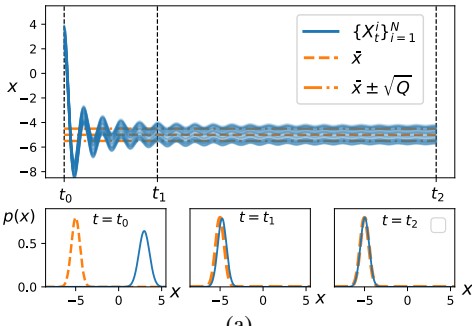 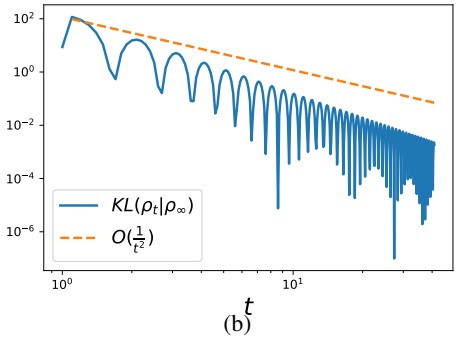

Figure 1: Simulation result for the Gaussian case (Example 4.1): (a) The time traces of the particles; (b) The KL-divergence as a function of time.

*approximation, the interaction term is approximated as*

$$\textit{(DE)} \quad I_t^{(N)}(X_t^i) = \frac{1}{2\epsilon} \frac{\sum_{j=1}^N g_\epsilon(X_t^i, X_t^j)(X_t^j - X_t^i)}{\sum_{j=1}^N g_\epsilon(X_t^i, X_t^j)} \tag{20}$$

*Despite the apparent similarity of the two formulae,* (19) *for diffusion map approximation and* (20) *for density estimation, the nature of the two approximations is different. The difference arises because the kernel $k_\epsilon(x, y)$ in* (19) *is data-dependent whereas the kernel in* (20) *is not. While both approximations are exact in the asymptotic limit as $N \uparrow \infty$ and $\epsilon \downarrow 0$, they exhibit different convergence rates. Numerical experiments presented in Figure 3(a)-(d) show that the diffusion map approximation has a much smaller variance for intermediate values of $N$. Theoretical understanding of the difference is the subject of continuing work.*

### 4.1 GAUSSIAN EXAMPLE

Consider the Gaussian example as described in Sec. 3.5. The simulation results for the scalar ($d = 1$) case with initial distribution $\rho_0 = \mathcal{N}(2, 4)$ and target distribution $\mathcal{N}(\bar{x}, Q)$ where $\bar{x} = -5.0$ and $Q = 0.25$ is depicted in Figure 1-(a)-(b). For this simulation, the numerical parameters are as follows: $N = 100$, $\phi_0(x) = 0.5(x - 2)$, $t_0 = 1$, $\Delta t = 0.1$, $p = 2$, $C = 0.625$, and $K = 400$. The result numerically verifies the $O(e^{-\beta_t}) = O(\frac{1}{t^2})$ convergence rate derived in Theorem 1 for the case where the target distribution is Gaussian.

### 4.2 NON-GAUSSIAN EXAMPLE

This example involves a non-Gaussian target distribution $\rho_\infty = \frac{1}{2}\mathcal{N}(-m, \sigma^2) + \frac{1}{2}\mathcal{N}(m, \sigma^2)$ which is a mixture of two one-dimensional Gaussians with $m = 2.0$ and $\sigma^2 = 0.8$. The simulation results are depicted in Figure 2-(a)-(b). The numerical parameters are same as in the Example 4.1. The interaction term is approximated using the diffusion map approximation with $\epsilon = 0.01$. The numerical result depicted in Figure 2-(a) show that the diffusion map algorithm converges to the mixture of Gaussian target distribution. The result depicted in Figure 2-(b) suggests that the convergence rate $O(e^{-\beta_t})$ also appears to hold for this non-log-concave target distribution. Theoretical justification of this is subject of continuing work.

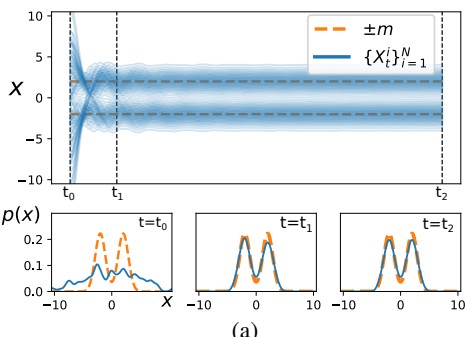 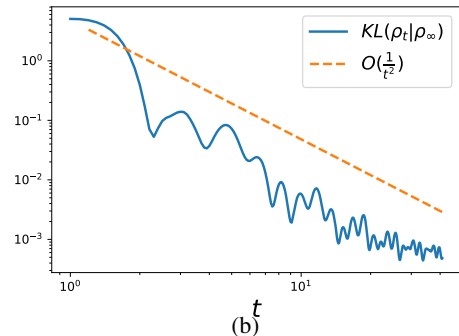

(a)                                                 (b)

Figure 2: Simulation result for the non-Gaussian case (Example 4.2): (a) The time traces of the particles; (b) The KL-divergence as a function of time.

## 4.3 COMPARISON WITH MCMC AND HMCMC

This section contains numerical experiment comparing the performance of the accelerated algorithm 1 using the diffusion map (DM) approximation (19) and the density estimation (DE)-based approximation (20) with the Markov chain Monte-Carlo (MCMC) algorithm studied in Durmus & Moulines (2016) and the Hamiltonian MCMC algorithm studied in Cheng et al. (2017).

We consider the problem setting of the mixture of Gaussians as in example 4.2. All algorithms are simulated with a fixed step-size of $\Delta t = 0.1$ for $K = 1000$ iterations. The performance is measured by computing the mean-squared error in estimating the expectation of the function $\psi(x) = x1_{x \geq 0}$. The mean-square error at the $k$-th iteration is computed by averaging the error over $M = 100$ runs:

$$\text{m.s.e}_k = \frac{1}{M} \sum_{m=1}^{M} \left( \frac{1}{N} \sum_{i=1}^{N} \psi(X_{t_k}^{i,m}) - \int \psi(x) \rho_\infty(x) \, dx \right)^2 \tag{21}$$

The numerical results are depicted in Figure 3. Figure 3(a) depicts the m.s.e as a function of $N$. It is observed that the accelerated algorithm 1 with the diffusion map approximation admits an order of magnitude better m.s.e for the same number of particles. It is also observed that the m.s.e decreases rapidly for intermediate values of $N$ before saturating for large values of $N$, where the bias term dominates (see discussion following Eq. 19).

Figure 3(b) depicts the m.s.e as a function of the number of iterations for a fixed number of particles $N = 100$. It is observed that the accelerated algorithm 1 displays the quickest convergence amongst the algorithms tested.

Figure 3(c) depicts the average computational time per iteration as a function of the number of samples $N$. The computational time of the diffusion map approximation scales as $O(N^2)$ because it involves computing a $N \times N$ matrix $[k_\epsilon(X^i, X^j)]_{i,j=1}^N$, while the computational cost of the MCMC and HMCMC algorithms scale as $O(N)$. The computational complexity may be improved by (i) exploiting the sparsity structure of the $N \times N$ matrix ; (ii) sub-sampling the particles in computing the empirical averages; (iii) adaptively updating the $N \times N$ matrix according to a certain error criteria.

Finally, we provide comparison between diffusion map approximation (20) and the density-based approximation (20): Figure 3(d) depicts the m.s.e for these two approximations as a function of the kernel-bandwidth $\epsilon$ for a fixed number of particles $N = 100$. For very large and for very small values

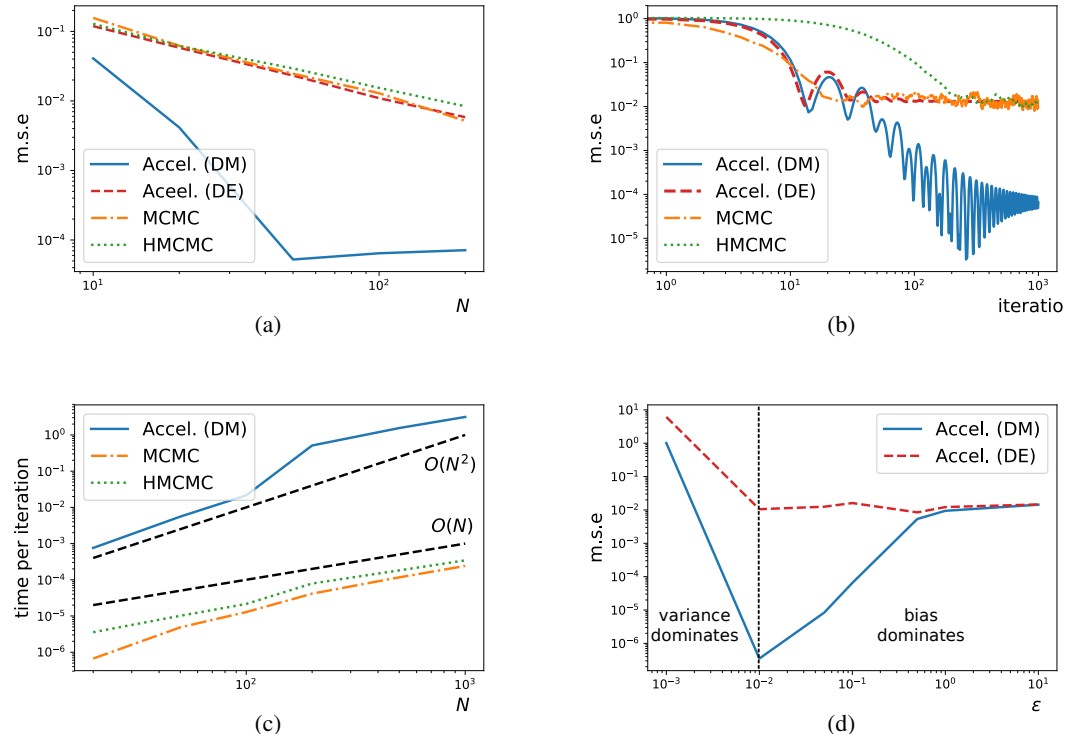

Figure 3: Simulation-based comparison of the performance of the accelerated algorithm 1 using the diffusion map (DM) approximation (19), the density estimation (DE)-based approximation (20) with the MCMC and HMCMC algorithms: (a) the mean-squared error (m.s.e) (21) as a function of the number of samples $N$; (b) the m.s.e as a function of the number of iterations; (c) the average computational time per iteration as a function of the number of samples; (d) m.s.e comparison between the diffusion map and the density estimation-based approaches as a function of the kernel bandwidth $\epsilon$.

of $\epsilon$, where bias and variance dominates the error, respectively, the two algorithms have similar m.s.e. However, for intermediate values of $\epsilon$, the diffusion map approximation has smaller variance, and thus lower m.s.e.

## 5 CONCLUSION AND DIRECTIONS FOR FUTURE WORK

The main contribution of this paper is to extend the variational formulation of Wibisono et al. (2016) to obtain theoretical results and numerical algorithms for accelerated gradient flow in the space of probability distributions. In continuous-time settings, bounds on convergence rate are derived based on a Lyapunov function argument. Two numerical algorithms based upon an interacting particle representation are presented and illustrated with examples. As has been the case in finite-dimensional settings, the theoretical framework is expected to be useful in this regard. Some direction for future include: (i) removing the technical assumption in the proof of the Theorem 1; (ii) analysis of the convergence under the weaker assumption that the target distribution satisfies only a spectral gap condition; and (iii) analysis of the numerical algorithms in the finite-$N$ and in the finite $\Delta t$ cases.

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

## A    PDE FORMULATION OF THE VARIATIONAL PROBLEM

An equivalent pde formulation is obtained by considering the stochastic optimal control problem (11) as a deterministic optimal control problem on the space of the probability distributions. Specifically, the process $\{\rho_t\}_{t\geq 0}$ is a deterministic process that takes values in $\mathcal{P}_{\mathrm{ac},2}(\mathbb{R}^d)$ and evolves according to the continuity equation

$$\frac{\partial \rho_t}{\partial t} = -\nabla \cdot (\rho_t u_t)$$

where $u_t : \mathbb{R}^d \to \mathbb{R}^d$ is now a time-varying vector field. The Lagrangian $\mathcal{L} : \mathbb{R}^+ \times \mathcal{P}_{\mathrm{ac},2}(\mathbb{R}^d) \times L^2(\mathbb{R}^d; \mathbb{R}^d) \to \mathbb{R}$ is defined as:

$$\mathcal{L}(t, \rho, u) := e^{\alpha_t + \gamma_t} \left[ \int_{\mathbb{R}^d} \frac{1}{2} |e^{-\alpha_t} u(x)|^2 \rho(x) \, \mathrm{d}x - e^{\beta_t} \mathsf{F}(\rho) \right] \tag{22}$$

The optimal control problem is:

$$\begin{aligned} \text{Minimize} \quad & \int_0^\infty \mathcal{L}(t, \rho_t, u_t) \, \mathrm{d}t \\ \text{Subject to} \quad & \frac{\partial \rho_t}{\partial t} + \nabla \cdot (\rho_t u_t) = 0 \end{aligned} \tag{23}$$

The Hamiltonian function $\mathcal{H} : \mathbb{R}^+ \times \mathcal{P}_{\mathrm{ac},2}(\mathbb{R}^d) \times \mathcal{C}(\mathbb{R}^d; \mathbb{R}) \times L^2(\mathbb{R}^d; \mathbb{R}^d) \to \mathbb{R}$ is

$$\mathcal{H}(t, \rho, \phi, u) := \langle \nabla \phi, u \rangle_{L^2(\rho)} - \mathcal{L}(t, \rho, u) \tag{24}$$

where $\phi \in \mathcal{C}(\mathbb{R}^d; \mathbb{R})$ is the dual variable and the inner-product $\langle \nabla \phi, u \rangle_{L^2(\rho)} := \int_{\mathbb{R}^d} \nabla \phi(x) \cdot u(x) \rho(x) \, \mathrm{d}x$

## B  RESTATEMENT OF THE MAIN RESULT AND ITS PROOF

We restate Theorem 1 below which now includes the pde formulation as well.

**Theorem 2.** *Consider the variational problem* (11)-(23).

(i) *For the probabilistic form* (11) *of the variational problem, the optimal control* $U_t^* = e^{\alpha_t - \gamma_t} Y_t$,
*where the optimal trajectory* $\{(X_t, Y_t)\}_{t \geq 0}$ *evolves according to the Hamilton's odes:*

$$\frac{\mathrm{d}X_t}{\mathrm{d}t} = U_t^* = e^{\alpha_t - \gamma_t} Y_t, \quad X_0 \sim \rho_0 \tag{25a}$$

$$\frac{\mathrm{d}Y_t}{\mathrm{d}t} = -e^{\alpha_t + \beta_t + \gamma_t} \nabla_\rho F(\rho_t)(X_t), \quad Y_0 = \nabla \phi_0(X_0) \tag{25b}$$

*where* $\phi_0$ *is a convex function, and* $\rho_t = Law(X_t)$.

(ii) *For the pde form* (23) *of the variational problem, the optimal control is* $u_t^* = e^{\alpha_t - \gamma_t} \nabla \phi_t(x)$,
*where the optimal trajectory* $\{(\rho_t, \phi_t)\}_{t \geq 0}$ *evolves according to the Hamilton's pdes:*

$$\frac{\partial \rho_t}{\partial t} = -\nabla \cdot (\rho_t \underbrace{e^{\alpha_t - \gamma_t} \nabla \phi_t}_{u_t^*}), \quad \text{initial condn.} \ \rho_0 \tag{26a}$$

$$\frac{\partial \phi_t}{\partial t} = -e^{\alpha_t - \gamma_t} \frac{|\nabla \phi_t|^2}{2} - e^{\alpha_t + \gamma_t + \beta_t} \nabla_\rho F(\rho) \tag{26b}$$

(iii) *The solutions of the two forms are equivalent in the following sense:*

$$Law(X_t) = \rho_t, \quad U_t = u_t(X_t), \quad Y_t = \nabla \phi_t(X_t)$$

(iv) *Suppose additionally that the functional* $F$ *is displacement convex and* $\rho_\infty$ *is its minimizer.*
*Define*

$$V(t) = \frac{1}{2} \mathsf{E}(|X_t + e^{-\gamma_t} Y_t - T_{\rho_t}^{\rho_\infty}(X_t)|^2) + e^{\beta_t}(F(\rho) - F(\rho_\infty)) \tag{27}$$

*where the map* $T_{\rho_t}^{\rho_\infty} : \mathbb{R}^d \to \mathbb{R}^d$ *is the optimal transport map from* $\rho_t$ *to* $\rho_\infty$. *Suppose also that*
*the following technical assumption holds:* $\mathsf{E}[(X_t + e^{-\gamma_t} Y_t - T_{\rho_t}^{\rho_\infty}(X_t)) \cdot \frac{\mathrm{d}}{\mathrm{d}t} T_{\rho_t}^{\rho_\infty}(X_t)] = 0$.
*Then* $\frac{\mathrm{d}V}{\mathrm{d}t}(t) \leq 0$. *Consequently, the following rate of convergence is obtained along the optimal*
*trajectory*

$$F(\rho_t) - F(\rho_\infty) \leq O(e^{-\beta_t}), \quad \forall t \geq 0$$

*Proof.* (i) The Hamiltonian function defined in (12) is equal to

$$\mathsf{H}(t, x, \rho, y, u) = y \cdot u - e^{\gamma_t - \alpha_t} \frac{1}{2} |u|^2 + e^{\alpha_t + \gamma_t \beta_t} \tilde{F}(\rho, x)$$

after inserting the formula for the Lagrangian. According to the maximum principle in prob-
abilistic form for (mean-field) optimal control problems (see (Carmona & Delarue, 2017,
Sec. 6.2.3)), the optimal control law $U_t^* = \arg\min_v \mathsf{H}(t, X_t, \rho_t, Y_t, v) = e^{\alpha_t - \gamma_t} Y_t$ and the
Hamilton's equations are

$$\frac{\mathrm{d}X_t}{\mathrm{d}t} = +\nabla_y \mathsf{H}(t, X_t, \rho_t, Y_t, U_t^*) = U_t^* = e^{\alpha_t - \gamma_t} Y_t$$

$$\frac{\mathrm{d}Y_t}{\mathrm{d}t} = -\nabla_x \mathsf{H}(t, X_t, \rho_t, Y_t, U_t^*) - \tilde{\mathsf{E}}[\nabla_\rho \mathsf{H}(t, \tilde{X}_t, \rho_t, \tilde{Y}_t, \tilde{U}_t^*)(X_t)]$$

where $\tilde{X}_t, \tilde{Y}_t, \tilde{U}_t^*$ are independent copies of $X_t, Y_t, U_t^*$. The derivatives

$$\nabla_x \mathsf{H}(t, x, \rho, y, u) = e^{\alpha_t + \beta_t + \gamma_t} \nabla_x \tilde{F}(\rho, x)$$

$$\nabla_\rho \mathsf{H}(t, x, \rho, y, u) = e^{\alpha_t + \beta_t + \gamma_t} \nabla_\rho \tilde{F}(\rho, x)$$

It follows that

$$\frac{\mathrm{d}Y_t}{\mathrm{d}t} = -e^{\alpha_t + \beta_t + \gamma_t} \left( \nabla_x \tilde{F}(\rho_t, X_t) + \tilde{\mathsf{E}}[\nabla_\rho \tilde{F}(\rho_t, \tilde{X}_t)(X_t)] \right) = -e^{\alpha_t + \beta_t + \gamma_t} \nabla_\rho \mathsf{F}(\rho)(X_t)$$

where we used the definition $\mathsf{F}(\rho) = \int \tilde{F}(x, \rho)\rho(x)\,\mathrm{d}x$ and the identity (Carmona & Delarue, 2017, Sec. 5.2.2 Example 3)

$$\nabla_\rho \mathsf{F}(\rho)(x) = \nabla_x \tilde{F}(\rho, x) + \int \nabla_\rho \tilde{F}(\rho, \tilde{x})(x)\rho(\tilde{x})\,\mathrm{d}\tilde{x}$$

(ii) The Hamiltonian function defined in (24) is equal to

$$\mathcal{H}(t, \rho, \phi, u) = \int \left[ \nabla\phi(x) \cdot u(x) - \frac{1}{2} e^{\gamma_t - \alpha_t} |u(x)|^2 \right] \rho(x)\,\mathrm{d}x + e^{\alpha_t + \gamma_t + \beta_t} \mathsf{F}(\rho)$$

after inserting the formula for the Lagrangian. According to the maximum principle for pde formulation of mean-field optimal control problems (see (Carmona & Delarue, 2017, Sec. 6.2.4)) the optimal control vector field is $u_t^* = \arg\min_v \mathcal{H}(t, \rho_t, \phi_t, v) = e^{\alpha_t - \gamma_t} \nabla\phi_t$ and the Hamilton's equations are:

$$\frac{\partial \rho_t}{\partial t} = +\frac{\partial \mathcal{H}}{\partial \phi}(t, \rho_t, \phi_t, u_t) = -\nabla \cdot (\rho_t \nabla u_t^*)$$

$$\frac{\partial \phi_t}{\partial t} = -\frac{\partial \mathcal{H}}{\partial \rho}(t, \rho_t, \phi_t, u_t) = -(\nabla\phi \cdot u^* - e^{\gamma_t - \alpha_t} \frac{1}{2}|u_t^*|^2 + e^{\alpha_t + \gamma_t + \beta_t} \frac{\partial \mathsf{F}}{\partial \rho}(\rho_t))$$

inserting the formula $u_t^* = e^{\alpha_t - \gamma_t} \nabla\phi_t$ concludes the result.

(iii) Consider the $(\rho_t, \phi_t)$ defined from (26). The distribution $\rho_t$ is identified with a stochastic process $\tilde{X}_t$ such that $\frac{\mathrm{d}\tilde{X}_t}{\mathrm{d}t} = e^{\alpha_t - \gamma_t} \nabla\phi_t(\tilde{X}_t)$ and $\mathrm{Law}(\tilde{X}_t) = \rho_t$. Then define $\tilde{Y}_t = \nabla\phi_t(\tilde{X}_t)$. Taking the time derivative shows that

$$\frac{\mathrm{d}\tilde{Y}_t}{\mathrm{d}t} = \frac{\mathrm{d}}{\mathrm{d}t} \nabla\phi_t(\tilde{X}_t) = \nabla^2\phi_t(\tilde{X}_t) \frac{\mathrm{d}\tilde{X}_t}{\mathrm{d}t} + \nabla\frac{\partial\phi_t}{\partial t}(X_t)$$

$$= e^{\alpha_t - \gamma_t} \nabla^2\phi_t(\tilde{X}_t)\nabla\phi_t(\tilde{X}_t) - e^{\alpha_t - \gamma_t} \nabla^2\phi_t(\tilde{X}_t)\nabla\phi_t(X_t) - e^{\alpha_t + \beta_t + \gamma_t} \nabla\frac{\partial\mathsf{F}}{\partial\rho}(\rho_t)(\tilde{X}_t)$$

$$= -e^{\alpha_t + \beta_t + \gamma_t} \nabla\frac{\partial\mathsf{F}}{\partial\rho}(\rho_t)(\tilde{X}_t)$$

$$= -e^{\alpha_t + \beta_t + \gamma_t} \nabla_\rho \mathsf{F}(\rho_t)(\tilde{X}_t)$$

with the initial condition $\tilde{Y}_0 = \nabla\phi_0(\tilde{X}_0)$, where we used the identity $\nabla_x \frac{\partial\mathsf{F}}{\partial\rho}(\rho) = \nabla_\rho \mathsf{F}(\rho)$ (Carmona & Delarue, 2017, Prop. 5.48). Therefore the equations for $\tilde{X}_t$ and $\tilde{Y}_t$ are identical. Hence one can identify $(X_t, Y_t)$ with $(\tilde{X}_t, \tilde{Y}_t)$.

(iv) The energy functional

$$V(t) = \underbrace{\frac{1}{2} \mathsf{E}\left[ |X_t + e^{-\gamma_t} Y_t - T_{\rho_t}^{\rho_\infty}(X_t)|^2 \right]}_{\text{first term}} + \underbrace{e^{\beta_t} (\mathsf{F}(\rho) - \mathsf{F}(\rho_\infty))}_{\text{second term}}$$

Then the derivative of the first term is

$$\mathsf{E}\left[ (X_t + e^{-\gamma_t} Y_t - T_{\rho_t}^{\rho_\infty}(X_t)) \cdot (e^{\alpha_t - \gamma_t} Y_t - \dot{\gamma}_t e^{-\gamma_t} Y_t - e^{\alpha_t + \beta_t} \nabla_\rho \mathsf{F}(\rho_t)(X_t) + \xi(T_{\rho_t}^{\rho_\infty}(X_t))) \right]$$

where $\xi(T_{\rho_t}^{\rho_\infty}(X_t)) := \frac{\mathrm{d}}{\mathrm{d}t} T_{\rho_t}^{\rho_\infty}(X_t)$. Using the scaling condition $\dot{\gamma}_t = e^{\alpha_t}$ the derivative of the first term simplifies to

$$\mathsf{E}\left[ (X_t + e^{-\gamma_t} Y_t - T_{\rho_t}^{\rho_\infty}(X_t)) \cdot (-e^{\alpha_t + \beta_t} \nabla_\rho \mathsf{F}(\rho_t)(X_t) + \xi(T_{\rho_t}^{\rho_\infty}(X_t))) \right]$$

Upon using the technical assumption, $\mathsf{E}[(X_t + e^{-\gamma_t}Y_t - T^{\rho_\infty}_{\rho_t}(X_t)) \cdot \xi(T^{\rho_\infty}_{\rho_t}(X_t))] = 0$ the derivative of the first term simplifies to

$$\mathsf{E}\left[(X_t + e^{-\gamma_t}Y_t - T^{\rho_\infty}_{\rho_t}(X_t)) \cdot (-e^{\alpha_t+\beta_t}\nabla_\rho F(\rho_t)(X_t))\right]$$

The derivative of the second term is

$$
\begin{aligned}
\frac{\mathrm{d}}{\mathrm{d}t}(\text{second term}) &= \dot{\beta}_t e^{\beta_t}(\mathsf{F}(\rho_t) - \mathsf{F}(\rho_\infty)) + e^{\beta_t}\frac{\mathrm{d}}{\mathrm{d}t}\mathsf{F}(\rho_t) \\
&= e^{\alpha_t+\beta_t}(\mathsf{F}(\rho_t) - \mathsf{F}(\rho_\infty)) + e^{\beta_t}\mathsf{E}[\nabla_\rho \mathsf{F}(\rho_t)(X_t)e^{\alpha_t-\gamma_t}Y_t]
\end{aligned}
$$

where we used the scaling condition $\dot{\beta}_t = e^{\alpha_t}$ and the chain-rule for the Wasserstein gradient (Ambrosio et al., 2008, Ch. 10, E. Chain rule). Adding the derivative of the first and second term yields:

$$\frac{\mathrm{d}V}{\mathrm{d}t}(t) = e^{\alpha_t+\beta_t}\left(\mathsf{F}(\rho_t) - \mathsf{F}(\rho_\infty) - \mathsf{E}\left[(X_t - T^{\rho_\infty}_{\rho_t}(X_t)) \cdot \nabla_\rho \mathsf{F}(\rho_t)(X_t)\right]\right)$$

which is negative by variational inequality characterization of the displacement convex function $\mathsf{F}(\rho)$ (Ambrosio et al., 2008, Eq. 10.1.7).

We expect that the technical assumption can be removed. This is the subject of the continuing work.

$\square$

## C    WASSERSTEIN GRADIENT AND GÂTEAUX DERIVATIVE

This section contains definitions of the Wasserstein gradient and Gâteaux derivative (Ambrosio et al., 2008; Carmona & Delarue, 2017).

Let $\mathsf{F} : \mathcal{P}_{\mathrm{ac},2}(\mathbb{R}^d) \to \mathbb{R}$ be a (smooth) functional on the space of probability distributions.

**Gâteaux derivative:** The Gâteaux derivative of $\mathsf{F}$ at $\rho \in \mathcal{P}_{\mathrm{ac},2}(\mathbb{R}^d)$ is a real-valued function on $\mathbb{R}^d$ denoted as $\frac{\partial \mathsf{F}}{\partial \rho}(\rho) : \mathbb{R}^d \to \mathbb{R}$. It is defined as a function that satisfies the identity

$$\left.\frac{\mathrm{d}}{\mathrm{d}t}\mathsf{F}(\rho_t)\right|_{t=0} = \int_{\mathbb{R}^d} \frac{\partial \mathsf{F}}{\partial \rho}(\rho)(x)(-\nabla \cdot (\rho(x)u(x)))\,\mathrm{d}x$$

for all path $\rho_t$ in $\mathcal{P}_{\mathrm{ac},2}(\mathbb{R}^d)$ such that $\frac{\partial \rho_t}{\partial t} = -\nabla \cdot (\rho_t u)$ with $\rho_0 = \rho \in \mathcal{P}_{\mathrm{ac},2}(\mathbb{R}^d)$.

**Wasserstein gradient:** The Wasserstein gradient of $\mathsf{F}$ at $\rho$ is a vector-field on $\mathbb{R}^d$ denoted as $\nabla_\rho \mathsf{F}(\rho) : \mathbb{R}^d \to \mathbb{R}^d$. It is defined as a vector-field that satisfies the identity

$$\left.\frac{\mathrm{d}}{\mathrm{d}t}\mathsf{F}(\rho_t)\right|_{t=0} = \int_{\mathbb{R}^d} \nabla_\rho \mathsf{F}(\rho)(x) \cdot u(x)\,\rho(x)\,\mathrm{d}x$$

for all path $\rho_t$ in $\mathcal{P}_{\mathrm{ac},2}(\mathbb{R}^d)$ such that $\frac{\partial \rho_t}{\partial t} = -\nabla \cdot (\rho_t u)$ with $\rho_0 = \rho \in \mathcal{P}_{\mathrm{ac},2}(\mathbb{R}^d)$.

The two definitions imply the following relationship (Carmona & Delarue, 2017, Prop. 5.48):

$$\nabla_\rho \mathsf{F}(\rho)(\cdot) = \nabla_x \frac{\partial \mathsf{F}}{\partial \rho}(\rho)(\cdot)$$

**Example:** Let $\mathsf{F}(\rho) = \int \log(\frac{\rho(x)}{\rho_\infty(x)})\rho(x)\,\mathrm{d}x$ be the relative entropy functional. Consider a path $\rho_t$ in $\mathcal{P}_{ac,2}(\mathbb{R}^d)$ such that $\frac{\partial \rho_t}{\partial t} = -\nabla \cdot (\rho_t u)$ with $\rho_0 = \rho \in \mathcal{P}_{ac,2}(\mathbb{R}^d)$. Then

$$\frac{\mathrm{d}}{\mathrm{d}t}\mathsf{F}(\rho_t) = \int \log(\frac{\rho_t(x)}{\rho_\infty(x)})\frac{\partial \rho_t}{\partial t}(x)\,\mathrm{d}x + \int \frac{\partial \rho_t}{\partial t}(x)\,\mathrm{d}x$$

$$= -\int \log(\frac{\rho_t(x)}{\rho_\infty(x)})\nabla \cdot (\rho_t(x)u(x))\,\mathrm{d}x$$

$$= \int \nabla_x \log(\frac{\rho_t(x)}{\rho_\infty(x)}) \cdot u(x)\,\rho_t(x)\,\mathrm{d}x$$

where the divergence theorem is used in the last step. The definitions of the Gâteaux derivative and Wasserstein gradient imply

$$\frac{\partial \mathsf{F}}{\partial \rho}(\rho)(x) = \log(\frac{\rho(x)}{\rho_\infty(x)})$$

$$\nabla_\rho \mathsf{F}(\rho)(x) = \nabla_x \log(\frac{\rho(x)}{\rho_\infty(x)})$$

# D    RELATIONSHIP WITH THE UNDER-DAMPED LANGEVIN EQUATION

A basic form of the under-damped (or second order) Langevin equation is given in Cheng et al. (2017)

$$\mathrm{d}X_t = v_t\,\mathrm{d}t$$
$$\mathrm{d}v_t = -\gamma v_t\,\mathrm{d}t - \nabla f(X_t)\,\mathrm{d}t + \sqrt{2}\,\mathrm{d}B_t \tag{28}$$

where $\{B_t\}_{t \geq 0}$ is the standard Brownian motion.

Consider next, the the accelerated flow (16). Denote $v_t := e^{\alpha_t - \gamma_t}Y_t$. Then, with an appropriate choice of scaling parameters (e.g. $\alpha_t = 0$, $\beta_t = 0$ and $\gamma_t = -\gamma t$ ):

$$\mathrm{d}X_t = v_t\,\mathrm{d}t$$
$$\mathrm{d}v_t = -\gamma v_t\,\mathrm{d}t - \nabla f(X_t)\,\mathrm{d}t - \nabla_x \log(\rho_t(X_t)) \tag{29}$$

The scaling parameters are chosen here for the sake of comparison and do not satisfy the ideal scaling conditions of Wibisono et al. (2016).

The sdes (28) and (29) are similar except that the stochastic term $\sqrt{2}\,\mathrm{d}B_t$ in (28) is replaced with a deterministic term $-\nabla_x \log(\rho_t(X_t))$ in (29). Because of this difference, the resulting distributions are different. Let $p_t(x, v)$ denote the joint distribution on $(X_t, v_t)$ of (28) and let $q_t(x, v)$ denote the joint distribution on $(X_t, v_t)$ of (29). Then the corresponding Fokker-Planck equations are:

$$\frac{\partial p}{\partial t}(x, v) = -\nabla_x \cdot (p_t(x, v)v) + \nabla_v \cdot (p_t(x, v)(\gamma v + \nabla f(x))) + \Delta_v p_t(x, v)$$

$$\frac{\partial q}{\partial t}(x, v) = -\nabla_x \cdot (q_t(x, v)v) + \nabla_v \cdot (q_t(x, v)(\gamma v + \nabla f(x))) + \nabla_v \cdot (q_t(x, y)\nabla_x \log(\rho_t(x)))$$

where $\rho_t(x) = \int q_t(x, v)\,\mathrm{d}v$ is the marginal of $q_t(x, y)$ on $x$. The final term in the Fokker-Planck equations are clearly different. The joint distributions are different as well.

The situation is in contrast to the first order Langevin equation, where the stochastic term $\sqrt{2}\,\mathrm{d}B_t$ and $-\nabla \log(\rho_t(X_t))$ are equivalent, in the sense that the resulting distributions have the same marginal distribution as a function of time. To illustrate this point, consider the following two forms of the Langevin equation:

$$\mathrm{d}X_t = -\nabla f(X_t)\,\mathrm{d}t + \sqrt{2}\,\mathrm{d}B_t \tag{30}$$
$$\mathrm{d}X_t = -\nabla f(X_t)\,\mathrm{d}t - \nabla \log(\rho_t(X_t)) \tag{31}$$

Let $p_t(x)$ denote the distribution of $X_t$ of (30) and let $q_t(x)$ denote the distribution of $X_t$ of (31). The corresponding Fokker-Planck equations are as follows

$$\frac{\partial p}{\partial t}(x) = -\nabla \cdot (p_t(x)\nabla f(x)) + \Delta p_t(x)$$

$$\frac{\partial q}{\partial t}(x) = -\nabla \cdot (q_t(x)\nabla f(x)) + \nabla \cdot (q_t(x)\nabla \log(\rho_t(x)))$$

$$= -\nabla \cdot (q_t(x)\nabla f(x)) + \nabla \cdot (q_t(x)\nabla \log(q_t(x)))$$

$$= -\nabla \cdot (q_t(x)\nabla f(x)) + \Delta q_t(x)$$

where we used $\rho_t(x) = q_t(x)$. In particular, this implies that the marginal probability distribution of the stochastic process $X_t$ are the same for first order Langevin sde (30) and (31) .

