# OpenReview forum: "Accelerated Gradient Flow for Probability Distributions"
_ICLR.cc/2019/Conference_

### Official Review · AnonReviewer3 · 2018-11-02
**Interesting extension of the Bregman Lagrangian framework, but quite expensive**

**Rating:** 6
**Confidence:** 4

**Review:**

Summary: This paper introduces a functional extension of the Bregman Lagrangian framework of Wibisono et al. 2016. The basic idea is to define accelerated gradient flows on the space of probability distribution. Because the defined flows include a term depending on the current distribution of the system, which is difficult to compute in general, the authors introduce an interacting particle approximation as a practical numerical approximation. The experiments are a proof-of-concept on simple illustrative toy examples.

Quality: The ideas are generally of high quality, but I think there might some typos (or at least some notation I did not understand). In particular
- tilde{F} is not defined for Table 1
- the lyapunov function for the vector column of table one includes a term referring to the functional over rho. I think this is a typo and should be f(x) - f(xmin) instead.

Clarity: The paper is generally clear throughout.

Originality & Significance: The paper is original to my knowledge, and a valuable extension to the interesting literature on the Bregman Lagrangian. The problem of simulating from probability distributions is an important one and this is an interesting connection between that problem and optimization.

Pros:
- An interesting extension that may fuel future study.

Cons:
- This algorithm appears naively to have an O(n^2) complexity per iteration, which is very expensive in terms of the number of particles. Most MCMC algorithms would have only O(n) complexity in the number of particles. This limits its applicability.

---

> ### Author Response · Authors · 2018-11-17
> **Response to reviewer 3**
>
> We thank the reviewer for reviewing the paper and for providing insightful comments.
>
>  “This algorithm appears naively to have an O(n^2) complexity per iteration, which is very expensive in terms of the number of particles.”
>
> This is an important criticism.  As part of comparison with MCMC, we have included figure-3-(c) which highlights the O(n^2) complexity of the proposed algorithm compared to O(n) complexity of MCMC.  In the revised version of the paper, we have now included text on the algorithm complexity and some approaches to ameliorate it: (i) exploiting the sparsity structure of the NxN matrix ; (ii) sub-sampling the particles in computing the empirical averages; (iii) adaptively updating the NxN matrix according to a certain error criteria.
>
> The notational issues in Table-1 have been fixed in the revised version of the paper.

---

### Official Review · AnonReviewer1 · 2018-11-04
**theoretically interesting**

**Rating:** 5
**Confidence:** 3

**Review:**

The articles adapt the framework developed in Wibisono & al to the (infinite dimensional) setting consisting in carrying out  gradient descent in the space of probability distributions.

PROS:
- the text is well written, with clear references to the literature and a high-level description of the current state-of-the-art.
- there is a good balance between mathematical details and high-level descriptions of the methods
- although I have not been able to check all the details of the proofs, the results appear to be correct.

CONS:
- while I think that this type of article is interesting, I was really frustrated to discover at the end that the proposed methods either rely on strong Gaussian assumptions, or  "density estimations". In other words, no "practical" method is really proposed.
- no comparison with other existing method is provided.

---

> ### Author Response · Authors · 2018-11-17
> **Response to reviewer 1**
>
> We thank the reviewer for reviewing the paper and for providing insightful comments.
>
> “No comparison with other existing method is provided.”
>
> The paper has been revised to now include  a comparison with the  MCMC and Hamiltonian MCMC algorithms.  The comparison is described in Sec 4.3 and the results of the comparison (accuracy and computational time) are depicted in Figure 3.
>
> “ … the proposed methods either rely on strong Gaussian assumptions or density estimation.”
>
> We would like to clarify that the proposed kernel algorithm does not involve explicit estimation of the density as an intermediate step.
> 1. We have included Remark 3 which clarifies the difference between the proposed kernel algorithm and an algorithm based on an explicit density estimation.
> 2. We have included results of numerical experiments comparing the kernel algorithm and the density estimation-based algorithm.  Results appear in Figure-3-(a)-(d) in Sec. 4.3.
> 3. In order to avoid the confusion with the density estimation, we now refer to the kernel approximation as the diffusion-map approximation.
>
> The algorithm based on Gaussian approximation is included because of its relationship to the Nesterov ode (see Remark 2).  Also, the algorithm may be useful in the cases where the density is unimodal (see the discussion following equation (18) in the paper).
>
> Finally, we note that the proposed form of the interaction term arises as a solution of the variational problem (which is the main contribution of our paper).  The theoretical results together with the positive preliminary numerical comparisons are likely to spur future work to develop more computationally efficient algorithms to approximate the interaction term.

---

### Official Review · AnonReviewer4 · 2018-11-17
**interesting derivation of 2nd gradient flows but with limited practical usefulness**

**Rating:** 4
**Confidence:** 4

**Review:**

This paper derives accelerated gradient flow formula in the space of probability measures from the view of optimal control formalism. The generalization of variational formulation from finite space to the space of probability measures seems new, but the resulting PDE seems to be a known result, which is the Fokker-Planck equation (with some minor modifications) for the 2nd order Langevin dynamic. From this point of view, the resulting algorithm from the derived PDE seems not having much practical advantage over SGHMC (a stochastic version of 2nd order Langevin dynamics).

Actually, I think the derivation of accelerated gradient flow formula from the view of optimal control formalism does not seem necessary. One can get the same formula by deriving it from Wasserstein gradient flows. When considering the functional as relative entropy, one can derive the formula simply from the Fokker-Planck equation of 2nd order Langevin dynamics. As a result, the proposed methods seems to be a new way to derive the Wasserstein gradient flow (or Fokker-Planck equation), which does not make impact the algorithm, e.g., both ways result in the same algorithm.

Besides, I found the writing needs to be improved. There are a lot of background missing, or the descriptions are not clear enough.  For example:
1. Page 2: the divergence operator is not defined, though I think it is a standard concept, but would be better to define it.
2. Page 2: the Wasserstein gradient and Gateaux derivative are not defined, what are the specific meanings of \nabla_\rho F(\rho) and \partial F / \partial \rho?
3. 1st line in Section 2: convex function f of d real variables seems odd, I guess the author means argument of f is d-dimensional variable.
4. Section 2, the authors directly start with the variational problem (3) without introducing the problem. Why do we need to variational problem? It would be hard to follow for some one who does not have such background.
5. Similarly, what is the role of Lyapunov function here in (6)? Why do we need it?
6. Why do you define the Lagrangian L in the form of (10)? What is the relation between (10) and (2)?
7. It is not clear what "The stochastic process (X_t, Y_t) is Gaussian" means in Proposition 1? It might need to be rephrased.
8. Second last line in page 5: I guess \nabla \log(\rho) should be \nabla\log(\rho_t).

For the theory, I think eq.15 only applies when the PDE, e.g. (13), is solved exactly, thus there is not too much practical impact, as it is well known from the Wasserstein gradient theory that the PDE decays exponentially, as stated in the theorem. When considering numerical solutions, I think this results is useless.

For the relation with SGHMC, let's look at eq.16. Actually, the derivative of the log term \nabla \log \rho_t(X_t)) is equivalent to a brownian motion term. This can be seen by considering the Fokker-Planck equation for Brownian motion, which is exactly d \rho_t = \Delta \rho_t. Consequently, instead of using the numerical approximations proposed later, one cane simply replacing this term with a Brownian motion term, which reduces to SGHMC (with some constant multipliers in front).

The authors then shows empirically that the proposed method is better than SGHMC, which I think only comes from the numerical methods.

For the kernel approximation, it makes the particles in the algorithm interactive. This resembles other particle optimization based algorithms such as SVGD, or the latest particle interactive SGLD proposed in [1] or [2[.  I think these methods need to be compared.

[1] Chen et al (2018), A Unified Particle-Optimization Framework for Scalable Bayesian Sampling.
[2] Liu et al (2018), https://arxiv.org/pdf/1807.01750.pdf

To sum up, though the derivation of accelerated gradient flow formula seems interesting, the resulting algorithm does not seem benefit from this derivation. The algorithm seems to be able to derived from a more direct way of using Wasserstein gradient flows, which results in a Wasserstein gradient flow for 2nd order Langevin dynamics, and is thus well known. The experiments are not convincing, and fail to show the advantage of the proposed method. The proposed method needs to be compared with other related methods.

---

> ### Author Response · Authors · 2018-11-18
> **Response to the reviewer 4**
>
> We thank the reviewer for reviewing the paper and for providing several helpful comments.
>
> “.. the resulting PDE seems to be a known result, which is the Fokker-Planck equation for the 2nd order Langevin dynamic.”
>
> It appears that the main concern of the reviewer is that the accelerated gradient flow proposed in our paper is the same as the second order Langevin equation or SGHMC?  We would like to clarify that this is not the case for the second order equation considered in this paper.
>
> For a first order Langevin equation, it is indeed true that the Brownian motion and $\nabla log(p)$ yield the same distribution. In other words, if one replaces the Brownian motion with $\nabla log(p)$ in the first order Langevin equation, the resulting Fokker-Planck equation and thus the distribution remains the same.
>
> However, this property does not hold for the second order Langevin equation considered in this paper. In the second order system, we are dealing with the joint distribution on position and momentum. If one replaces the Brownian motion (in the momentum update) with $\nabla log(p)$ where $p$ is the marginal on the position, the resulting Fokker-Planck equations are different.  Consequently, the distributions are also different.
>
> Since this is an important point, we have included a new section (Appendix D) as part of the supplementary material in the paper to show the difference between the first order and the second order cases.
>
> “.. Actually, I think the derivation of accelerated gradient flow formula from the view of optimal control formalism does not seem necessary ..”
>
> Variational formulation of fundamental equations is a cornerstone of Mathematics.
>     1. Lagrangian mechanics is a variational formulation of Newtonian mechanics;
>     2. Feynman’s path integral formulation of quantum mechanics;
>     3. For the Fokker-Planck equation, the celebrated gradient flow construction of the Jordan- Kinderlehrer-Otto;
>     4. Finally, Wibisono et. al. is itself a variational formulation of the Nesterov ode (which has been known since 1980-s).
> `    5. As noted in the introduction, the objective of this paper is to generalize Wibisono el. al. (2016). So a variational construction is natural.
>
> In all these cases 1-4, variational formulations have been worthy of study not only for numerical reasons but also because of their rich mathematical structure, geometric aspects which makes the derivation of models and algorithms independent of the choice of coordinates, first integrals and Lyapunov function which provides insights into conserved quantities and convergence analysis etc.  Variational formulations have also been useful for numerics, e.g., development of symplectic integrators.
>
> “ … One can get the same formula by deriving it from Wasserstein gradient flows. ”
>
>  We disagree that the proposed accelerated algorithm can be derived using Wasserstein gradient flows. Or at least, we are not aware of how to do that.
>
> " ... though the derivation of accelerated gradient flow formula seems interesting, the resulting algorithm does not seem benefit from this derivation"
>
> The concern of the reviewer is justified. The numerical algorithm is obtained from discretizing the Hamilton’s equations (16).  These equations are directly derived from the variational formulation. One may try to obtain the numerical algorithm directly from the variational formulation by discretizing (in both time and space) the Lagrangian directly. For example, the symplectic integration is the result of such a time discretization. We believe that it is possible express the variational problem in terms of particles in the Gaussian setting with the solution given by the proposed numerical algorithm in the Gaussian settings. However, doing so in more general setting is beyond the scope of this paper.
>
> “The authors then shows empirically that the proposed method is better than SGHMC, which I think only comes from the numerical methods.”
>
> Regarding the numerical comparison, the revised version of the paper includes comparison to MCMC, HMCMC (which is the same as the second order Langevin equation), and a method based on the density estimation. Please note that, as clearly described in the Introduction, the main contribution of the paper is the variational formulation and the generalization of the Wibisono et. al. and not the numerical algorithm in of itself. The numerical experiments are included to illustrate the theoretical results (e.g., accelerated convergence rates), show the potential and limitations of the proposed algorithm (e.g., bias-variance tradeoff depicted in Fig. 3 (d) and computational complexity depicted in Fig. 3 (c), and provide some preliminary comparisons with MCMC and HMCMC (Fig. 3).
>
> We do not claim that the proposed algorithm is better than all the existing algorithms.  Such a claim will require extensive numerical experiments which are outside the scope of this paper.

---

> ### Author Response · Authors · 2018-11-18
> **Response to the reviewer 4 (part 2)**
>
> The reviewer also suggested several improvements as part of an enumerated list 1-8.  These suggestions have been incorporated in the revised version of the paper:
>
> 1) The definition of the divergence is indeed standard but now appears as part of Notation (on page 2).
>
> 2) We have added a new section Appendix C as part of the Supplementary material.  The definition of the Wasserstein gradient and Gateaux derivative appears as part of this section.
>
> 3) The sentence has now been rephrased to avoid confusion.
>
> 4) We do not completely understand the reviewer’s concern. The variational formulation in the finite-dimensional Euclidean setting is due to Wibisono et al. (2016).  The motivation for the same appears in the Introduction.
>
> 5) The Lyapunov function is useful to obtain convergence results.
>
> 6) The definition of the Lagrangian (10) is a core contribution of this paper.  The proposed definition represents a generalization of the Lagrangian (2) proposed by Wibisono et. al.  The relationship between the two is summarized in Table I, discussed in Introduction.  Additional relationship appears in Prop. 1 where it is shown that we recover the continuous limit of Nesterov ode in the Gaussian setting. Furthermore, the result of Theorem.1-(ii) shows that one also obtains the same convergence rate as in Wibisono, et. al. (2017).
>
> 7) The text now reads “the stochastic process (X_t,Y_t) is a Gaussian process”. The definition of a Gaussian process is standard.
>
> 8) The typo has been fixed in the revised version of the paper.

---

> ### Comment · AnonReviewer4 · 2018-11-23
> **thanks for the clarification**
>
> I think the revision is getting better. However, I still don't quite get how to go from an ODE/PDE to the Lagrangian. What is the relation between these two as well as the  Lyapuno function. I think some write of this part is necessary.
>
> “ … One can get the same formula by deriving it from Wasserstein gradient flows. ”
> One straightforward way to directly solve the Wasserstein gradient flow with particle approximation, like what was done in this paper: A blob method for diffusion.
>
> Although the paper considers a different PDE (part of it is the first order Langevin PDE), I think same technique can be applied to the 2nd order PDE, which can also be defined by a Wasserstein gradient flow on the joint space.
>
> So what I said is this paper seems to proposes a different way to get the 2nd order Langevin PDE, but the final numerical solution seems to be a numerical solution directly from some specific Wasserstein gradient flow. I tend to keep my decision.

---

> ### Author Response · Authors · 2018-11-26
> **Good discussion**
>
> We thank the reviewer for reading our response and the revised version of the paper. We think this discussion is helpful and important in understanding the paper.
>
> “… I still don't quite get how to go from an ODE/PDE to the Lagrangian. What is the relation between these two as well as the Lyapuno function ...”
>
> It is actually the other way around. First, the Lagrangian is formulated. Then, ODE/PDEs are obtained from the Lagrangian.
>
> The Lagrangian function in Eq. (2) is motivated by Lagrangian mechanics. In Lagrangian mechanics, the Lagrangian function is equal to the difference between kinetic energy and the potential energy. The Lagrangian in Eq. (2) has similar form when one think of X_t as position, U_t as velocity, and the objective function f(x) as potential energy. The time-varying scaling parameters are employed to obtain the convergence to the minimum.
>
> Lyaponov functions are commonly understood as functions that capture the “energy” of the system. They should be positive everywhere, except at the equilibrium, where they should equal to zero. The Lyapunov function Eq. (6) that appears in the paper has almost the form of sum of the kinetic energy and the potential energy.
>
> Please note that we point out the role of “kinetic energy” and “potential energy” in the Lagrangian in Eq. (2). For detailed discussion, we refer the interested reader to Wibinoso et. al. (2016).
>
>  “… Although the paper considers a different PDE (part of it is the first order Langevin PDE), I think same technique can be applied to the 2nd order PDE, which can also be defined by a Wasserstein gradient flow on the joint space.“
>
> The accelerated flow proposed in our paper can NOT be defined by a Wasserstein gradient flow on the joint space. It does not belong to the family of Wasserstein gradient flows that appear in the paper that you mentioned [A blob method for diffusion] for any choice of functionals.
>
> The situation is similar to the vector case. The accelerated flow is not a gradient flow. It can NOT be obtained from gradient of any function on the joint space. It is actually a Hamiltonian flow as it was shown in Wibinoso et. al. (2016), also reviewed in Sec. 2 of our paper.
>
> Similarly, the accelerated flow for probability distribution that is presented in our paper is NOT a Wasserstein gradient flow with respect to any functional.  It does not belong to the family of the Wasserstein gradient flows for any choice of the functionals. Actually, the accelerated flow presented in our paper is a Hamiltonian flow on the space of probability distributions, as it is shown in Theorem 1.
>
> In fact, if it was possible, showing that the second order Langevin equation is a Wassertein gradient flow is a great contribution, and we are very interested to know how this derivation is done exactly.
>
> “this paper seems to proposes a different way to get the 2nd order Langevin PDE, but the final numerical solution seems to be a numerical solution directly from some specific Wasserstein gradient flow”
>
> We disagree with the conclusion for the following reasons:
>
> 1) The accelerated flow presented in our paper is different from the 2nd order Langevin equation both in PDE form and in ODE form (please see our previous response and Appendix D).
>
> 2) This paper is proposing a variational formulation to construct accelerated flow for probability distributions. We want to emphasize the important role of variational formulation (please see our previous response). Note that the same criticism also holds for Lagrangian mechanics. It can also be said that Lagrangian mechanics is “a different way” to obtain Newton’s second law. However, such evaluation is undermining the importance of Lagrangian mechanics.
>
> 3) The numerical algorithm is obtained from discretizing the accelerated flow which itself is obtained from the variational formulation. The numerical algorithm is not obtained from any Wasserstein gradient flow because the accelerated flow proposed in our paper is not a Wasserstein gradient flow.

---

### Author Response · Authors · 2018-11-17
**Summary of responses**

We thank the reviewers for carefully reading the paper and for providing helpful comments.  Both the reviewers agreed that the problem is important, the contributions are original, and the paper is well written.  Broadly, the reviewers raised two concerns on the numerical aspects of the paper:

Concern 1: Lack of comparison with existing methods.

Concern 2: Complexity/practicality of the proposed algorithm.

Our answers to these top-level concerns are as follows:

Answer to concern 1: The paper has been revised to now include also a comparison with the state-of-the-art Markov Chain Monte-Carlo (MCMC) and Hamiltonian MCMC algorithms.

Answer to concern 2: The main contribution of this paper is theoretical. The preliminary numerical results demonstrate that, using the same number of samples, the proposed numerical algorithm achieves better accuracy compared to the state-of-the-art.  The theoretical contributions together with these preliminary results are likely to fuel future study to develop more practical lower-complexity algorithms.  Additional details appear in our response to the reviewers.

---

### Meta-Review · Area_Chair1 · 2018-12-14
**Interesting ideas but method is not practical**

**Confidence:** 5
**Recommendation:** Reject

**Metareview:**

This paper developed an accelerated gradient flow in the space of probability measures. Unfortunately, the reviewers think the practical usefulness of the proposed approach is not sufficiently supported by realistic experiments, and the clarity of the paper need to be significantly improved. The authors' rebuttal resolved some of the confusion the reviewers had, but we believe further substantial improvement will make this work a much stronger contribution.